**communications** engineering

# Economically viable geological CO$_2$ storage from direct air capture has critical threshold of 70% CO$_2$ concentration
Le Zhang ⓘ , Yunfeng Liang ⓘ ✉, Arata Kioka ⓘ & Takeshi Tsuji ⓘ ✉

Direct Air Capture (DAC), a key component of Carbon Capture and Storage (CCS), has been widely studied. However, its large-scale deployment is hindered by the high energy cost of purifying captured CO$_2$. Using impure CO$_2$ can reduce energy consumption and overall costs, but it also lowers storage efficiency. This work employs molecular dynamics simulations to examine storage efficiency by analyzing the impurity systems' density across a wide temperature and pressure range. The results indicate a strong similarity between the density changes at the macroscopic level and the Van der Waals interaction changes at the molecular level. Additionally, the Normalized Storage Efficiency caused by Impurities (NSEI) is proposed, which can be used for storage potential and cost evaluation. A detailed NSEI analysis suggests that CO$_2$ concentration should reach at least 70% to achieve economically viable storage. This finding provides practical guidance for DAC capture system design and impurity CCS project planning.

In recent years, Carbon Capture and Storage (CCS) technology has been extensively studied, not only for its potential to mitigate environmental issues caused by rising CO$_2$ levels[1–5], but also for its contribution to enhancing oil and gas recovery[6–8]. Currently, widely recognized CCS technologies include bioenergy with CCS (BECCS) and Direct Air Capture (DAC). However, BECCS faces limitations such as long investment period, land use, and water resource requirements, making it less viable for addressing the increasingly urgent issue of CO$_2$ emissions in the short term[9]. It was assessed that up to 5% of 2018 European emissions could be mitigated through BECCS under current tech and economic conditions[10]. This is far from sufficient to achieve net-zero emissions. DAC technology is emerging as a critical solution on the path to net-zero emissions. By capturing CO$_2$ directly from the air and permanently storing it, DAC provides a means to offset emissions that are challenging to eliminate, such as those from long-distance transport and heavy industry, while also addressing legacy emissions. According to the IEA Net Zero Emissions by 2050 Scenario, DAC technologies are projected to capture more than 85 Mt CO$_2$ by 2030 and around 980 Mt CO$_2$ by 2050, requiring a dramatic and accelerated scale-up from the current level of almost 0.01 Mt CO$_2$ captured annually[11].

The main DAC technologies currently in use can be categorized into three types: solid-based DAC (s-DAC), liquid-based DAC (l-DAC), and emerging DAC technologies[11,12]. s-DAC captures CO$_2$ by using solid adsorbents in an adsorption-desorption process, where CO$_2$ is passed through the adsorbent. Recently, an amine-functionalized COF-999 (Covalent Organic Frameworks, COF) achieved over 80 mol% CO$_2$ capture

efficiency from the air[13]. l-DAC employs a selective liquid-phase chemical absorption of atmospheric CO$_2$, yielding a CO$_2$ stream with >95 mol% purity[14]. However, energy consumption is still a concern. Emerging DAC technologies include membrane-based DAC (m-DAC), which is widely studied due to its smaller footprint and simpler setup and operation. Although m-DAC remains in its early stages, it consumes only 1/30 to 1/20 of the energy required by amine-based adsorption systems[15]. As a purely physical process with no toxic emissions, m-DAC minimizes environmental impact and public concerns. Moreover, in regions with advanced renewable energy infrastructure, integrating renewables with m-DAC further lowers energy consumption[16]. Fujikawa et al. reported that a four-stage membrane system achieved a CO$_2$ permeate concentration of 42.4%[17]. New developing membrane materials, such as graphene oxide-supported deep eutectic solvent membranes[18], also exhibit excellent separation performance. The low atmospheric CO$_2$ concentration demands high energy input and extra processing[11], making cost the main barrier to large-scale DAC deployment. Even under optimistic point-source capture conditions, such as capturing directly from industrial chimneys with CO$_2$ concentrations of ~20%, the energy requirements remain substantial[19]—let alone the additional costs associated with adsorbent degradation caused by impurities such as sulfur and nitrogen compounds.

Among the three DAC technologies, l-DAC is relatively more cost-effective due to its technological maturity and availability of raw materials, estimated at USD 94–232/t CO$_2$, though still exceeding the industry target of USD 100/t CO$_2$. In comparison, s-DAC costs range from USD 100–600/t

School of Engineering, The University of Tokyo, Tokyo, Japan. ✉e-mail: liang@sys.t.u-tokyo.ac.jp; tsuji@sys.t.u-tokyo.ac.jp

$CO_2$. On the other hand, storage costs are comparatively lower. Most onshore projects have storage costs below USD 10/t $CO_2$[11]. Furthermore, the potential for $CO_2$ storage is vast. Globally, the estimated storage capacity is as much as 10,000–30,000 Gt $CO_2$[3]. Even in seismically active Japan, over 100 Gt of $CO_2$ could be stored—enough to cover more than 100 years of the country's emissions[20]. The relatively low storage costs combined with the vast storage potential make the storage of impure $CO_2$ a feasible strategy for reducing overall costs.

Previous research on DAC has primarily focused on capture methods and absorbents. For instance, an earlier study reported the glass-like phase transformation issues of new water-lean $CO_2$ capture solvents during $CO_2$ capture[21]. Roberto reviewed ongoing research and attempts at applying membrane separation for DAC[19]. In contrast, studies on storage efficiency have largely focused on pure $CO_2$. Bachu reviewed the assessment of $CO_2$ storage efficiency in deep saline aquifers, highlighting the influence of storage site characteristics, confining aquifers, operating conditions, and regulatory frameworks[22]. Li et al. investigated the impact of pressure buildup on storage efficiency during pure $CO_2$ injection[23]. Kim et al. developed an artificial neural network model trained on various geological factors and their ranges, significantly improving the prediction accuracy for $CO_2$ storage performance[24]. Regarding the storage of impure $CO_2$, Wang et al. studied the density of impure $CO_2$ using both equation-of-state models and experimental approaches, pointing out that the amount of the reduced $CO_2$ storage capacity induced by impurities is not necessarily proportional to their molar fractions. Furthermore, they found that the introduction of impurities could even enhance $CO_2$ storage capacity depending on the critical temperature of the impurities[25]. While the equation-of-state method accurately predicts pure $CO_2$ density in the gaseous phase, its predictions for impure $CO_2$ or liquid and supercritical states are less reliable. The use of Molecular Dynamics (MD) simulations has become a widely reported approach for studying CCS issues[26-29]. For example, Liang et al. utilized MD simulations to discover that wettability exhibits different changes under pressure for different mineral surfaces, contributing to a better understanding of the geological storage process[27]. Xie et al. investigated the effects of impurities such as $N_2$ and $O_2$ on $CO_2$ using MD simulations, with their research primarily focusing on the behavior of supercritical $CO_2$[28]. Tsuji et al. used MD to investigate the density differences of $CO_2$ mixtures captured by m-DAC with $N_2$ and $O_2$ impurities[29]. They found that introducing

small amounts of air impurities (~20 mol%) could double storage costs compared to 99 mol% $CO_2$. Despite the doubled cost in storage, considering the storage cost is far below the purification cost, injecting impure $CO_2$ may be economically feasible. However, the study's temperature and pressure range are limited, and the cost model does not account for a cost breakdown.

This paper applies MD simulation methods to analyze the influence of impurities from DAC on $CO_2$ storage over a wide range of temperatures and pressures (20–120 °C, 60–320 bar) encompassing most of CCS projects globally. This work begins by analyzing $CO_2$ density in impure systems. The results indicate that $O_2$ and $N_2$ introduced from DAC have a nonlinear negative impact on $CO_2$ density in impure systems. Molecular-level investigation finds that Van der Waals interactions between molecules correlate with density variations. Based on the density calculation, the Normalized Storage Efficiency caused by Impurities (NSEI) is proposed to evaluate the storage potential and cost in impure $CO_2$ projects. A nonlinear decline in NSEI is observed, with the most pronounced drop occurring at 70 mol% $CO_2$ concentration. This suggests that in impure $CO_2$ storage projects, a minimum $CO_2$ concentration of 70 mol% is necessary for better utilization of storage pore space. From the capture side, achieving at least 70 mol% $CO_2$ capture concentration provides a clear target for optimizing the capture system and material design. As a result, 70 mol% $CO_2$ concentration is identified as an economically viable concentration. The NSEI analysis of several well-known CCS projects validates this conclusion, further highlighting the practical implications of economically viable concentration.

## Results
### Density profile on a wide P-T range
At first, the densities of pure $CO_2$, $N_2$, and $O_2$ were calculated to validate our results from MD simulations. As shown in Fig. 1, the simulation results align closely with NIST WebBook data[30], with an average error below 1%. Furthermore, pure $CO_2$ density was also calculated using the Peng–Robinson equation-of-state (P–R EoS) model[31,32]. The P–R EoS aligns well with NIST data at low pressures but shows deviations at higher pressures. These results confirm the reliability of our simulation methodology and force field for accurate density predictions.

DAC captures low-purity $CO_2$ directly from the air, with $O_2$ and $N_2$ being the primary impurities. Among the three main DAC technologies, membrane-based DAC (m-DAC) offers energy advantages. It is reported that the $CO_2$ produced in a four-stage membrane system could reach 42.4 mol%, while also achieving net-negative $CO_2$ emissions—meaning the $CO_2$ emitted during the purification process is less than the amount captured. A slight improvement in membrane selectivity may increase the $CO_2$ concentration in the permeate to 50% or higher[17]. This work focuses on examining the impact of storing impure $CO_2$ with concentrations ranging from 50 mol% to 100 mol%. A survey of several well-known CCS projects worldwide—CarbFix[33,34], Cranfield[35], Sleipner[36-38], Decatur[39,40], and Nagaoka[41,42], and China Offshore[23]—guided the temperature (20–120 °C) and pressure (60–320 bar) ranges in this study. For details, refer to the Methods section.

Density heatmap results across a wide range of temperature and pressure for various $CO_2$ concentrations were shown in Fig. 2. In Fig. 2a, the boundary between the liquid and gas phases is evident. In the supercritical region, the Widom line is observable, reflecting the reasonableness of the MD simulations[43]. Additionally, it can be observed that as $O_2$ and $N_2$ increase, this boundary gradually shifts to the right, indicating a progressive reduction in $CO_2$ density. Within the studied temperature and pressure range, the highest density of 982.67 kg $m^{-3}$ occurs at 20 °C and 320 bar in the bottom-right corner of the plot **a** 100 mol% $CO_2$. Figure 2b–g, compared to Fig. 2a demonstrate that as $O_2$ and $N_2$ are increasingly mixed with $CO_2$, the $CO_2$ density decreases under the same temperature and pressure conditions. This is because, as $N_2$ and $O_2$ are introduced, the system occupies a larger volume, reducing $CO_2$ density.

As temperature increases and pressure decreases, the mixture's density decreases. At 220 bars (Pressure in the Decatur project), Fig. 3a shows how

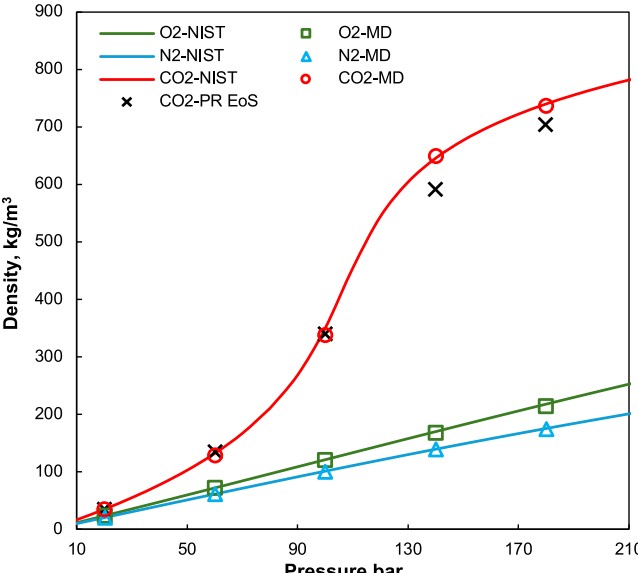

**Fig. 1 | Density validation results for pure molecular system at 325.65 K.** Molecular dynamics simulation results matched well with NIST data for $N_2$, $O_2$, and $CO_2$. P–R EoS model is employed for pure $CO_2$ density prediction, while deviations at higher pressures were observed.

**Fig. 2 | CO₂ Density heatmap at various concentrations (Unit: kg m⁻³). a** 100% CO₂; **b** 95% CO₂; **c** 90% CO₂; **d** 80% CO₂; **e** 70% CO₂; **f** 60% CO₂; **g** 50% CO₂. Supercritical region and critical point of pure CO₂ were plotted in each panel as reference.

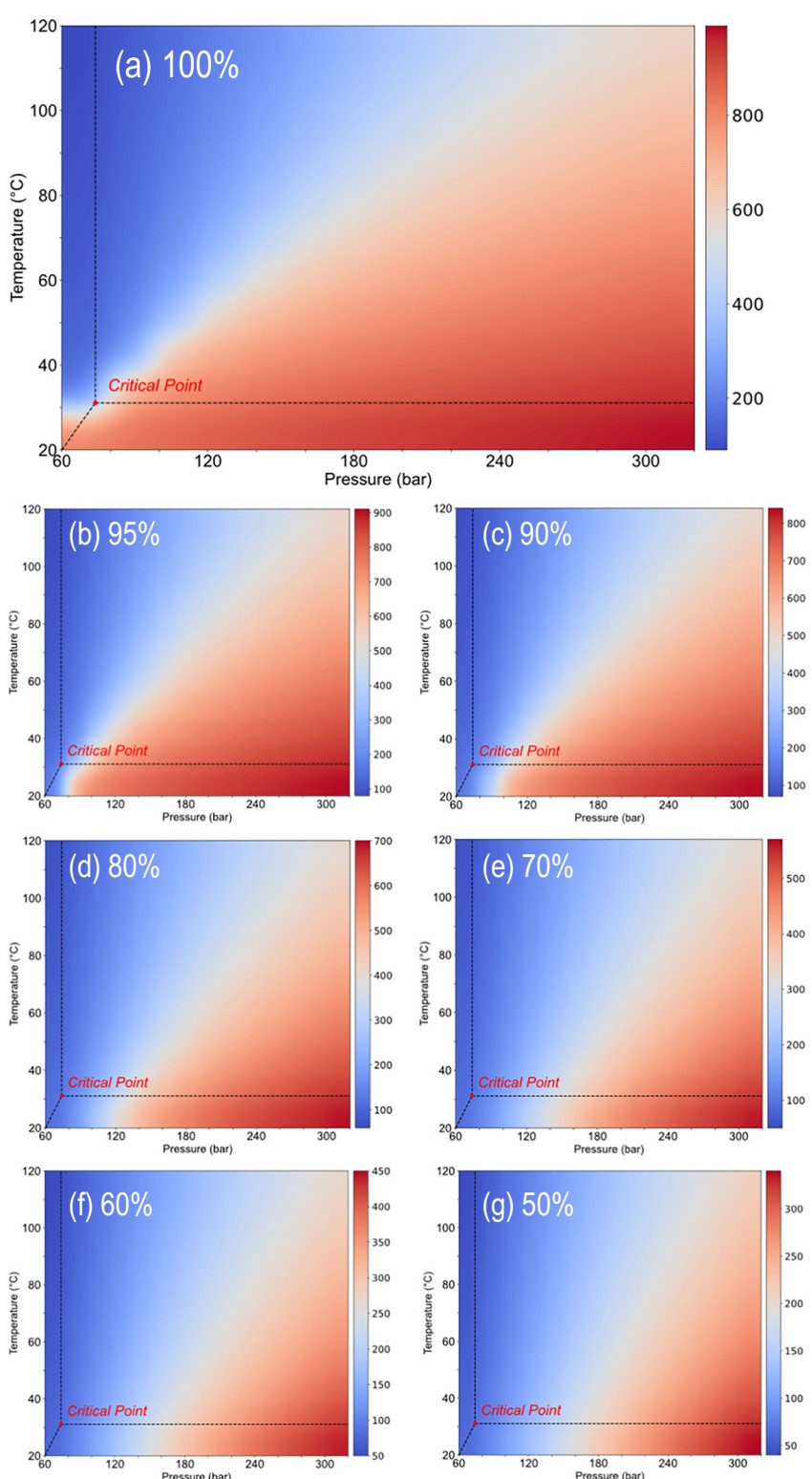

CO₂ density varies with temperature at different purity levels. As the temperature rises, the density of the mixture gradually decreases, and this decreasing trend becomes less pronounced at higher temperatures. In other words, the density difference (density difference between pure CO₂ and 50 mol% CO₂) gradually narrows as the temperature increases. In Fig. 3a, the density range at 20 °C (labeled b in Fig. 3a) is only 41% of the range at 120 °C (a in Fig. 3a). Figure 3b, illustrates the variation in density with pressure at 50 °C, corresponding to the storage temperature in the Decatur

and Nagaoka projects. It can be observed that at 100 mol% CO₂, there is a noticeable density increase near 100 bars, which is distinct from the behavior observed at other concentrations. This behavior occurs because the critical temperatures of N₂ and O₂ are substantially lower than that of CO₂. According to the Mixing rule (determined by the molar fractions of components in a mixture), the critical temperature of the system decreases rapidly with the addition of N₂ and O₂. For example, at 80 mol% CO₂, the system's critical temperature is −3.44 °C, while at 50 mol% CO₂, the critical

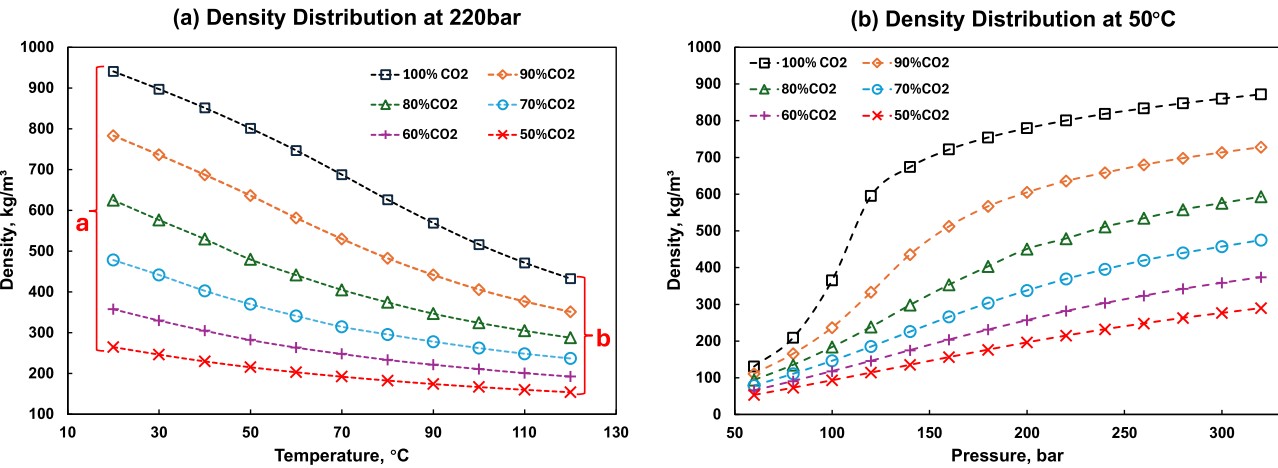

**Fig. 3 | Density profile for various $CO_2$ concentrations. a** Density as a function of temperature at 220 bars and **b** as a function of pressure at 50 °C. In (**b**), a sharp density increase is observed around 100 bars for pure $CO_2$, which is expected due to the crossing of the Widom line.

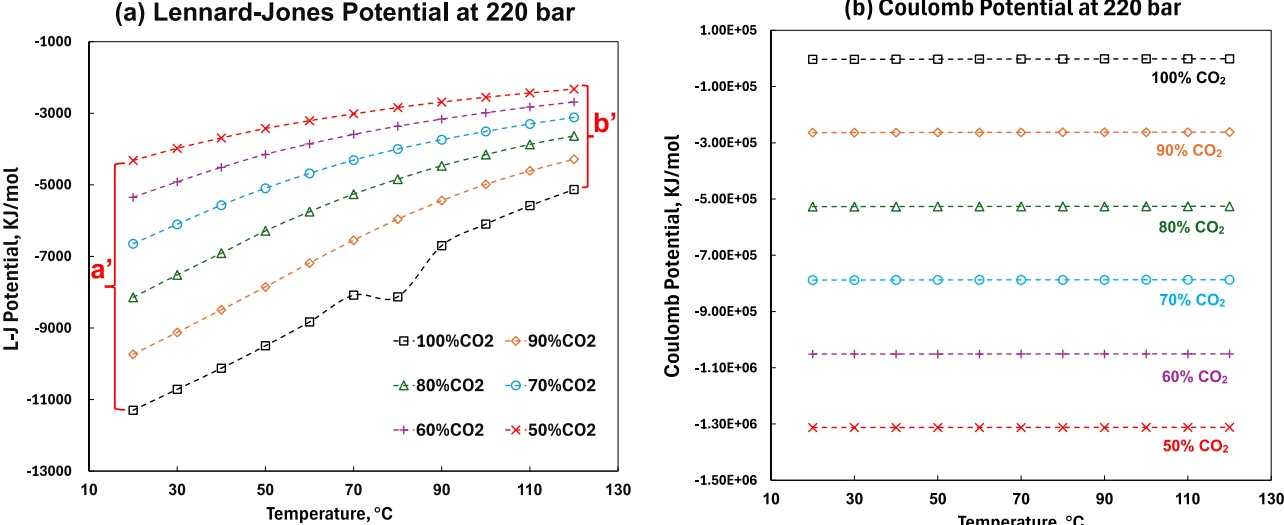

**Fig. 4 | Lennard-Jones potential and Coulomb potential at 220 bars as a function of temperature at various $CO_2$ concentrations. a** Displays the difference in Lennard-Jones potential, which correlates well with the observed density variations. **b** Shows the Coulomb potential is almost constant, where temperature has a minimal impact.

temperature drops to −55.1 °C. As the system moves further from 50 °C, the compressibility transitions across the critical point become less pronounced, resulting in relatively smoother curves.

The Lennard-Jones (LJ) and Coulomb potential were further analyzed. This result accounts for both the short-range interactions and the reciprocal space contributions of the LJ and Coulomb potential. The negative LJ potential in Fig. 4a indicates that intermolecular interactions are predominantly attractive, that is, mainly van der Waals interactions. The larger the absolute value of the LJ potential, the stronger the intermolecular attraction. It can be observed that systems with lower $CO_2$ concentrations have smaller absolute LJ potential values compared to pure $CO_2$. This is because the addition of $O_2$ and $N_2$ disrupts the gathering behavior of $CO_2$ molecules, reducing the overall attractive forces in the system, which is reflected in the weakened attractive term of the LJ potential. Additionally, the LJ potential profile exhibits a similar trend of larger potential ranges at low temperatures and smaller ranges at high temperatures. To further investigate this, we calculated the potential difference at high and low temperatures (labeled respectively as b' and a' in Fig. 4a), with the ratio being 40.2%, which closely aligns with the density difference ratio. The similarity between energy and density trends suggests that this phenomenon is likely driven by the effect of temperature on intermolecular interactions. In the pure $CO_2$ system, molecular interactions are more sensitive to temperature changes, whereas the addition of $O_2$ and $N_2$ impurities will mitigate. Compared with the LJ potential, the Coulomb potential in Fig. 4b is almost constant. The Coulomb potential is little affected by temperature, which reinforces the use of the LJ potential (here, mainly van der Waals interactions) to explain the influence of temperature on density changes. Radial Distribution Function (RDF) and coordination number analysis results of impure systems are shown in Supplementary Fig. 1. Detailed discussion can be found in Supplementary Note 1.

**Normalized Storage Efficiency Caused by Impurities (NSEI)**

In impure systems, the reduction in $CO_2$ density directly leads to a decrease in $CO_2$ storage capacity. Currently, storage capacity evaluation is primarily conducted using two approaches: the storage mechanism-based method and the volumetric method. The former assesses total storage potential by considering all trapping mechanisms—for example, saline aquifers typically involve four primary mechanisms: structural, dissolution, mineral, and residual trapping[22]. This approach requires detailed geological descriptions, reservoir dynamics, and comprehensive numerical simulations. In contrast, the volumetric method focuses solely on $CO_2$ storage efficiency ($E$), defined as the ratio of accessible $CO_2$ volume ($V_{CO_2}$) to total pore volume ($V_{Total}$).

This approach allows for assessing $CO_2$ storage capacity by evaluating $CO_2$ intrusion into the pore space[44,45].

$$M_{CO_2} = A \cdot h \cdot \varphi \cdot \rho_{CO_2} \cdot E_{geo} \qquad (1)$$

where $M_{CO_2}$ is the storage potential (kg), $A$ is the area of the storage region ($m^2$), $h$ is the thickness of the formation (m), $\varphi$ is the porosity, $\rho_{CO_2}$ is density of $CO_2$ (kg m$^{-3}$), and $E_{geo}$ is storage efficiency factor.

Impurities influence on impure $CO_2$ storage is primarily reflected in the reduction of $CO_2$ density. Building on Tsuji et al. finding that injecting impure $CO_2$ increases costs[29], this study adopts their storage cost model to calculate the Normalized Storage Efficiency caused by Impurities (NSEI) and evaluate the impact on storage efficiency and costs. The calculation of NSEI follows:

$$NSEI = \rho_i / \rho_p \qquad (2)$$

which compares the density of impure $CO_2$ ($\rho_i$) with that of pure $CO_2$ ($\rho_p$) under various temperature and pressure conditions. By using NSEI, the influence of impurities on storage efficiency can be normalized. Through the modification of $CO_2$ density in the volumetric calculation method, it streamlines the assessment process for storage efficiency factors in scenarios involving the injection of impure $CO_2$. This metric enables us to quantify the impact of impurities on storage efficiency. NSEI can also be used to estimate the cost associated with impure $CO_2$ storage. This cost evaluation assumed that the storage cost is related to the amount of $CO_2$ within the allocated pore space. When the density of $CO_2$ is higher, we can store a large amount of $CO_2$ in the allocated pore space. Regarding other capture-related costs, such as pressurization and transportation, an impure system requires more energy, and the associated costs can be expressed as 1/NSEI times those of pure $CO_2$. For example, when NSEI is 0.7, the associated impure $CO_2$ storage cost would be approximately $1 \div 0.7 \approx 1.43$ times that of pure $CO_2$.

Figure 5 illustrates the NSEI under different conditions. The figure highlights a distinct low-efficiency region in the bottom-left corner, corresponding to conditions near the critical state of $CO_2$. Under these temperature and pressure conditions, storage efficiency is significantly reduced due to the susceptibility of $CO_2$ to phase transitions, which result in substantial volumetric changes. This low-efficiency region is primarily concentrated around the critical point of $CO_2$, and its range gradually expands as $CO_2$ purity decreases. Furthermore, six CCS projects were plotted in Fig. 5 based on their respective temperature and pressure conditions. For the CarbFix project, its average temperature of 26.5 °C was selected. Differences in temperature and pressure among projects lead to varying NSEI values, even at the same $CO_2$ concentration. For example, at 50 mol% concentration, the NSEI for the CarbFix and Cranfield projects approaches 35%, whereas the efficiency for the Sleipner project falls below 15%. From a cost perspective, this implies that under the same 50 mol% condition, the NSEI-derived storage cost at CarbFix and Cranfield is about 2.3 times that at Sleipner. This underscores the sensitivity of NSEI and associated cost to $CO_2$ concentrations.

## Effective storage based on cost analysis

One of the major challenges hindering the large-scale adoption of DAC technology is its high cost. According to a DAC report by the International Energy Agency (IEA) in 2024, the capture cost for solid-based DAC (s-DAC) can reach up to USD 540 per ton of $CO_2$, whereas liquid-based DAC (l-DAC) has a lower capture cost of up to USD 340 per ton[11]. Under current technological conditions, l-DAC demonstrates a cost advantage over s-DAC. Carbon Engineering (CE) has been developing an aqueous DAC(l-DAC) system since 2009, and after 15 years of stable development and operation, its energy and cost analyses are considered representative. In this system, with an inlet $CO_2$ concentration of 600 ppm (higher than the atmospheric level of 415 ppm) and pure $CO_2$ delivered at 15 MPa, capturing

one ton of $CO_2$ requires either 8.81 GJ of natural gas or 5.25 GJ of natural gas combined with 366 kWh of electricity[14]. The levelized costs are estimated to be in the range of USD 94–232 per ton of $CO_2$, which aligns with IEA's projected cost range. In comparison, s-DAC systems have an estimated cost range of USD 100–600 per ton of $CO_2$[11,16]. The emerging m-DAC technology, due to its lack of large-scale capture experience, presents challenges in estimating costs. However, it is anticipated that its costs will primarily consist of membrane prices, energy consumption, and operational labor costs. In a reported four-stage m-DAC simulation, energy consumption and membrane requirements decreased progressively from the first to the fourth stage. The first stage requires approximately 11.6 kWh per kg-$CO_2$ per day, while the fourth stage consumes only 1/60 of the energy required for the first stage[17]. This reduction occurs because, as more membranes are added, the partial pressure of separated $CO_2$ decreases, thereby lowering the energy required for vacuuming. According to the IEA, under current technological conditions, the cost of $CO_2$ capture via DAC for large-scale applications (1 Mt $CO_2$ per year) is estimated to range between USD 125−335 per ton $CO_2$, depending on the type of capture technology, specific plant configurations, and electricity and energy prices. From a storage perspective, costs mainly consist of seismic operations, well drilling and completion, monitoring, etc. Compared to DAC technology, these processes benefit from extensive validation in the oil and gas industry, making their costs relatively stable and reasonable. The IEA reports that storage costs can be as low as USD 10 per ton in the United States, with approximately half of its offshore storage estimated to be available at costs below USD 35 per ton of $CO_2$.

Given the high costs associated with capturing $CO_2$, injecting impure $CO_2$ offers a potential reduction in the materials and operational costs of CCS projects, which can help lower overall expenditures. Even compared to the ideal market price of USD 100 per ton, storage costs typically account for only about 30% of the total. Injecting $CO_2$ at 50% concentration would increase costs by approximately USD 30, but this is offset by considerable savings from reduced energy consumption during purification, extended material lifespans, and decreased operations for maintenance, far exceeding the additional USD 30 cost. Studying the injection of impure $CO_2$ holds the potential to further reduce overall CCS costs. Based on the above considerations, we set NSEI = 0.5 as the threshold storage efficiency value. When NSEI equals 0.5, it indicates that the storage cost is exactly twice that of storing pure $CO_2$. The area where NSEI > 0.5 is further defined as the Effective Storage Area. The proportion of the Effective Storage Area decreases as $CO_2$ concentration declines. Figure 6 illustrates the proportion of this region (i.e., the Effective Storage Area) relative to the total storage region as a function of $CO_2$ concentration, with the red dotted curve representing the variation under global conditions. It is evident that the proportion of the effective storage region increases rapidly between $CO_2$ concentrations of 60 and 80 mol% with showing an inflection point at ~70 mol%. This suggests that on a global scale, achieving a minimum $CO_2$ concentration of 70% is crucial to fully utilizing $CO_2$ storage potential.

## Discussion

To provide more practical advice for impure $CO_2$ storage, this study further analyzes the temperature and pressure gradients of potential storage sites, which can be found in Supplementary Table 1. Detailed description can be found in Supplementary Note 2. Temperature gradients across various CCS sites are relatively stable, generally around 30 °C km$^{-1}$, except for the Decatur project, which has a lower gradient of 18.2 °C km$^{-1}$. Studies on the global geothermal gradient distribution in sedimentary basins suggest that the average geothermal gradient is 36.1 °C km$^{-1}$, with a standard deviation of 27.3 °C per km[46]. The pressure gradient data indicate that most projects have a pressure gradient of around 100 bar km$^{-1}$, while China's shale gas projects typically reach 150 bar km$^{-1}$, likely due to the greater burial depth of shale formations in China. Based on these observations, this study establishes the P-T relationship as follows:

$$T - T_0 = (K_T / K_P) \times P \qquad (3)$$

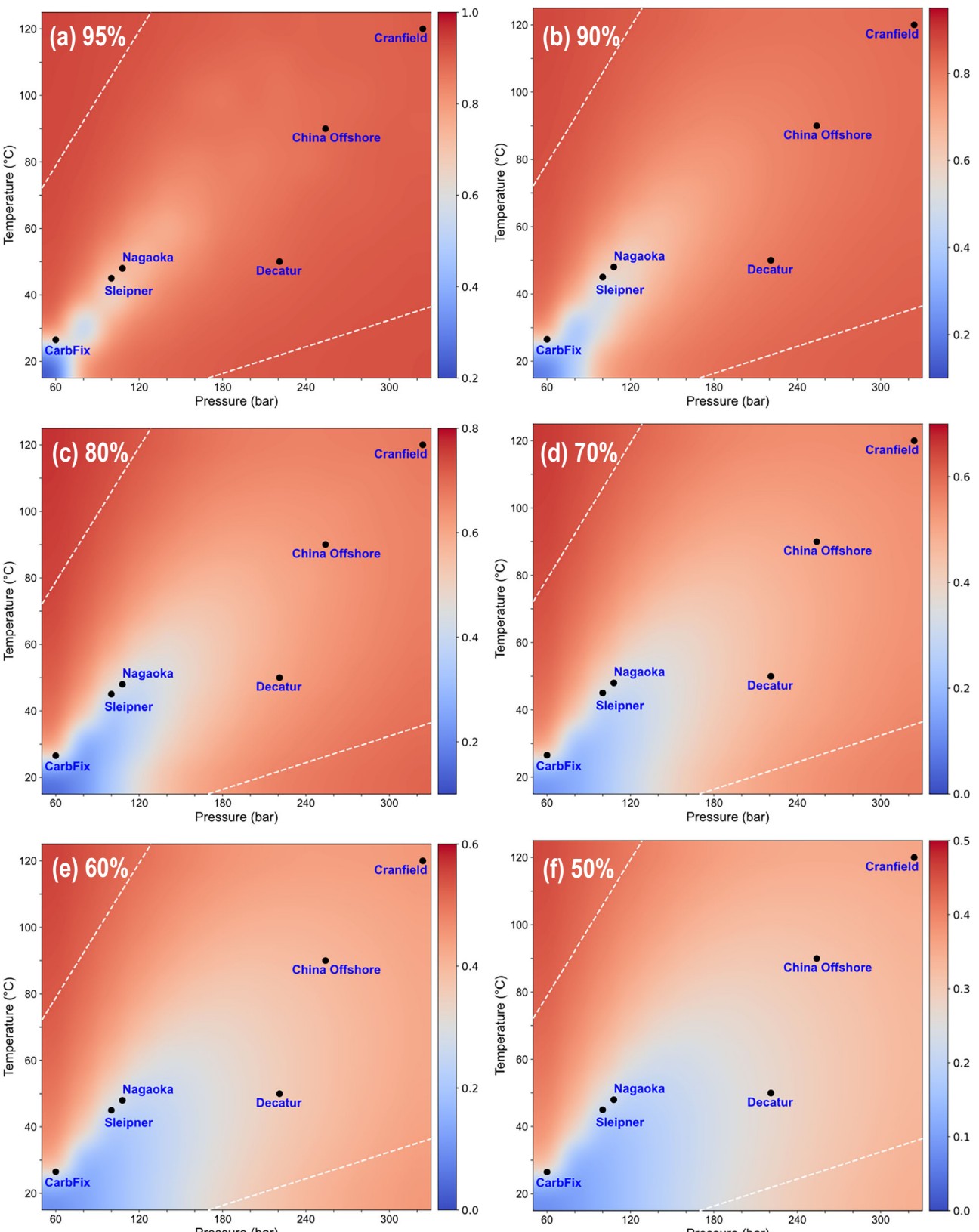

**Fig. 5 | Normalized Storage Efficiency Caused by Impurities (NSEI) at different pressures and temperatures. a** 95% $CO_2$; **b** 90% $CO_2$; **c** 80% $CO_2$; **d** 70% $CO_2$; **e** 60% $CO_2$; **f** 50% $CO_2$. The white dash line represents inaccessible temperature-pressure boundaries. Pressure and temperature are impractical out of these boundaries. Blue area (low NSEI area) expands as the $CO_2$ concentration decreases. The six reference CCS projects exhibit different NSEI values, owing to their distinct pressure and temperature conditions.

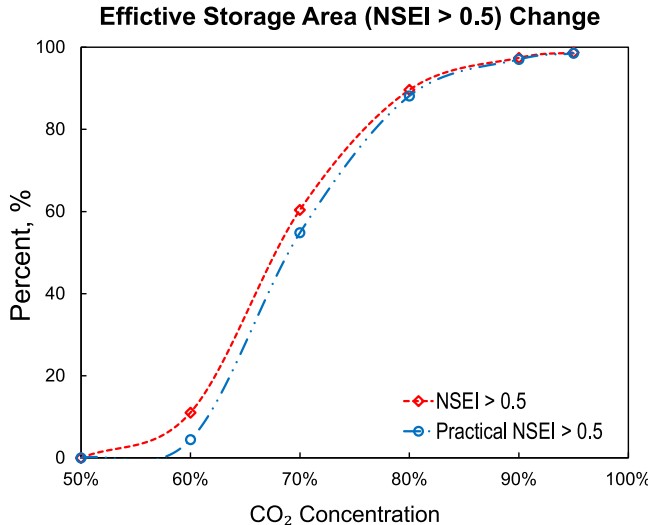

**Fig. 6 | Change of effective storage area (NSEI > 0.5) with CO$_2$ concentration.** Both overall NSEI and practical NSEI show an inflection point at ~70%, suggesting this concentration as the critical threshold for economically viable geological storage.

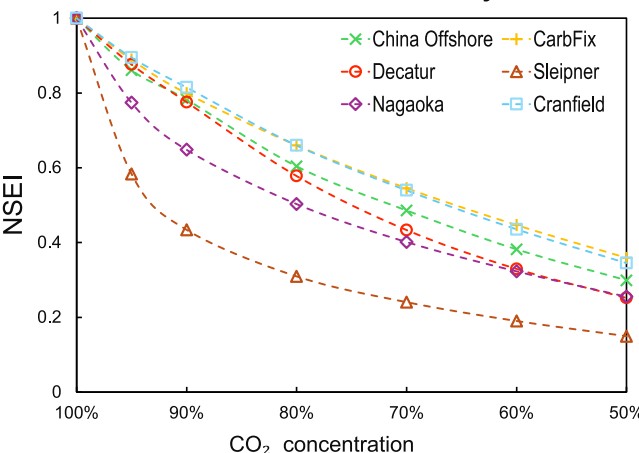

**Fig. 7 | Normalized Storage Efficiency caused by Impurities (NSEI) for different projects.** The Cranfield and CarbFix projects exhibit higher NSEI values under all examined pressure–temperature conditions, reflecting their favorable P–T environments. While the Sleipner site has the lowest NSEI and is therefore not recommended for impure CO$_2$ injection and storage.

where $T_0$ is the initial temperature, $K_T$ is the temperature gradient (43.34 ± 27.3 °C km$^{-1}$), and $K_P$ is the pressure gradient (90–160 bar km$^{-1}$). For offshore projects, the seabed temperature can be as low as 3 °C, while for onshore projects, the initial temperature is set at 25 °C.

Using the P-T relationship, two inaccessible temperature-pressure boundaries were established, as shown by the white dashed lines in Fig. 5. These two boundaries define the limits of feasible temperature-pressure conditions, separating inaccessible areas in the NSEI plot. The area between these lines reflects the realistic temperature conditions expected in CCS projects. In Fig. 6, the blue dotted curve represents the variation of Effective Storage Area with CO$_2$ concentration, after excluding the inaccessible P-T range. At 70% CO$_2$ concentration, the graph also exhibits the steepest slope, which is consistent with the results obtained under global conditions. This finding provides valuable concentration guidance for impure CCS projects, indicating that increasing CO$_2$ purity above 70% is necessary to enhance NSEI more effectively.

Additionally, it is worth noting that beyond the temperature and pressure gradients listed in this work, some notable CCS projects were not included in our references. For example, the CarbFix2 project, an upscaled version of the CarbFix project, operates at higher temperatures and pressures[47–49]. According to reports, at the CO$_2$ storage site (~2000 m depth), the temperature can reach 220–260 °C due to its proximity to the Hellisheidi geothermal power plant (1.5 km away). The Hengill volcanic system produces a profound geothermal influence that is fundamentally different from that of the sedimentary basins studied here. Moreover, the subsurface rocks in the CarbFix2 project primarily consist of olivine tholeiitic basalts, with chlorite, epidote, and calcite being the most common secondary minerals. These conditions provide a favorable environment for CO$_2$ and H$_2$S mineral reactions, in contrast to the sandstone- and shale-dominated sedimentary basins, where basalt content is relatively low. Another key distinction is that that CarbFix2's fluid mass is unstable due to continuous extraction for power generation. This differs from oil and gas reservoirs and common saline aquifer CCS projects, which typically require a relatively stable environment to minimize leakage risks. However, if the unique temperature-pressure conditions of the CarbFix2 project were included in the potential storage site analysis, all the temperature and pressure conditions considered in this work would still fall within the accessible area. Therefore, the conclusion that a 70% CO$_2$ concentration is an economically viable threshold remains valid.

We also conducted a comparison of NSEI variations across referenced projects, which could provide a practical guide for implementing impure CO$_2$ storage in these projects. The variation in NSEI across 6 projects under varying impurity conditions is clearly illustrated in Fig. 7. Due to differences in temperature and pressure conditions, each project exhibits a distinct NSEI. Notably, CarbFix and Cranfield demonstrate relatively high NSEI for impure CO$_2$, whereas Sleipner shows the lowest NSEI. This disparity is attributed to Sleipner's proximity to the low-efficiency region identified in the bottom-left corner. Additionally, the Sleipner and Nagaoka projects both exhibits pronounced changes in NSEI at 95%, which aligns with the previous analysis of the effective storage region. Note that in CarbFix and CarbFix2 projects, the CO$_2$-saturated water phase was injected, and CO$_2$ was stored by carbon mineralization. However, it is also possible to inject a supercritical CO$_2$-rich phase into these reservoirs, which can likely save water usage and lead to a higher mineralization rate[50]. The cost mentioned above would then mean a cost for the injection of CO$_2$ with impurities.

In summary, this study aims to explore a balance between CO$_2$ purity and cost, striving to maximize storage efficiency while keeping the cost within acceptable limits. To this end, it sets a target of 70% as the economically viable concentration to be reached in the near future in the capture community, especially for the DAC techniques. A material system with high selectivity or specificity would be desirable to effectively increase CO$_2$ concentration. On the other hand, it also attracts attention in the storage community, for impurity CCS project planning. When implementing impure CCS projects, more factors—such as engineering and additional thermodynamics parameters—should be incorporated for a well-informed decision. Moreover, there are efforts to utilize CO$_2$ from DAC, for example, to convert CO$_2$ to value-added chemicals/products, which are not considered in this study. Beyond that, the influences of water, sulfur-, and nitrogen-oxides from the industrial sector on impure CO$_2$ storage, as well as the potential impact on CO$_2$ utilization and conversions, still require further investigation. Overall, the information that we learned from this study is essential for DAC capture technologies development and storage project planning.

## Methods
### Simulation details
All Molecule Dynamics (MD) simulations were performed with GROMACS package v2024.4[51]. The particle mesh Ewald (PME) method was applied for both long-range electrostatic and Lennard-Jones interactions with a Fourier spacing of 0.12 nm and a cutoff distance of 1.4 nm[52,53]. The

## Table 1 | The non-bond parameters for molecules

| Molecules | Atom | $\sigma(nm)$ | $\varepsilon(KJ/mol)$ | $q(e)$ | Critical Temperature(K) TraPPE-model / NIST | Critical Pressure (Bar) TraPPE-model / NIST |
|---|---|---|---|---|---|---|
| $CO_2$ | C | 0.2800 | 0.2244918 | 0.700 | 306.2 / 304.21 | 77.7 / 73.843 |
| | O | 0.3050 | 0.6568464 | −0.350 | | |
| $N_2$ | N | 0.3310 | 0.2993040 | −0.482 | 126.5 / 126.19 | 34.6 / 33.98 |
| | MW | 0.0000 | 0.0000000 | 0.964 | | |
| $O_2$ | O | 0.3020 | 0.4073860 | −0.113 | 153.5 / 154.58 | 50.63 / 51.1 |
| | MW | 0.0000 | 0.0000000 | 0.226 | | |

## Table 2 | Typical $CO_2$ storage site worldwide

| Field Project and Location | Temperature (°C) | Pressure (Bar) |
|---|---|---|
| CarbFix, Iceland | 20–33 | ~ 60 |
| Cranfield, USA | 120 | 324 |
| Nagaoka, Japan | 48 | 108 |
| Sleipner, Norway | 45 | 100 |
| Decatur IBDP, USA | 50 | 221 |
| China Offshore, China | ~90 | 254 |

bonds were constrained using a linear constraint solver (LINCS) algorithm with bond constraints added[54]. Additionally, 3-D periodic boundary conditions (PBC) were applied to all simulations. Because the DAC captures $CO_2$ directly from the air, we considered the $N_2$ and $O_2$ as the primary impurities. Water is not considered since it is inevitable in $CO_2$ storage, which is not specific for $CO_2$ from DAC only, but for $CO_2$ from all types of sources. The ratio of $N_2$ to $O_2$ in the simulation is kept at 4:1. Each system contains 1380 molecules. For systems with different $CO_2$ concentrations, corresponding amounts of $CO_2$ were replaced with $N_2$ and $O_2$ in the same ratio. For example, under the condition of 50 mol%, the system contains 690 $CO_2$ molecules, 552 $N_2$ molecules, and 138 $O_2$ molecules. A total of seven typical cases are considered: 100% (pure $CO_2$), 95 mol%, 90 mol%, 80 mol %, 70 mol%, 60 mol%, and 50 mol%. A $10 \times 10 \times 10$ $nm^3$ box was prepared for holding molecules at the beginning stage. The system firstly relaxed through steepest descent algorithm (Energy minimization). After energy minimization, 1-ns NPT simulation with velocity-rescaling thermostat[55] and Berendsen barostat[56] was performed for equilibrium purpose. After that, 5-ns NPT simulation with Nosé-Hoover thermostat and an isotropic Parrinello-Rahman barostat was performed for production[57–59]. Time step was set as 1 fs, so there are 1,000,000 steps for equilibrium and 5,000,000 steps for production runs. Only the last 2 million steps of the final 5 million production simulation steps were used for result analysis, thereby avoiding any early-stage disequilibrium.

### Force fields used for molecules
The field force used for $CO_2$, $N_2$, and $O_2$ were from TraPPE-small (Transferable Potential for Phase Equilibria), which is built to calculate small but vital molecules by the University of Minnesota[60,61]. These force fields were used by various studies, showing excellent agreement in density and vapor-liquid equilibria calculation[62–64]. All Molecules built in TraPPE-small force field have rigid structures (i.e., no bonded potential) and do not otherwise fit into another TraPPE family. Detailed non-bonded parameters and partial charge can be found in Table 1. The Lorentz—Berthelot combination rules were employed to calculate the interactions among molecules. The density calculation uses the isothermal—isobaric (NPT) ensemble to validate the selected force fields for impurities.

### Pressure and temperature selection
Investigations are conducted into several representative CCS projects worldwide, as can be found in Table 2. The CarbFix project is a model for $CO_2$ mineralization research. Reportedly, the injected $CO_2$ reacts with basalt, achieving permanent disposal within just two years[33,34]. The Cranfield project, located in Mississippi, USA, features high temperature and pressure conditions[35]. Sleipner[36–38] and Decatur[39,40] are among the world's earliest CCS projects, whose designs and processes have served as references for many subsequent projects. Nagaoka is the first Japanese pilot project for $CO_2$ geological storage[41,42]. The China Offshore project is one of the country's latest CCS initiatives, with massive storage potential[23]. From a geographical perspective, the selected six CCS projects span major global economies and diverse regions, making them highly representative. These well-known CCS projects help identify a temperature range of 20 to 120 °C and a pressure range of 60 to 320 bars for this study. These conditions span the critical point of $CO_2$, making the findings relevant not only for storage but also for $CO_2$ pipeline transport.

## Data availability
Initial and final configuration of molecular dynamics simulation at each concentration, along with the simulation settings, can be found in the Supplementary Data Set File (in.zip file). All Data will be made available on request.

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

## Acknowledgements

The authors acknowledge the Japan Society for the Promotion of Science (JSPS) for a Grant-in-Aid for Scientific Research (21H05202, 22H05108, 22K03927, 23K04647, and 24H00440) and ENEOS Xplora Inc. for their financial support. L.Z. is grateful for a scholarship from the Graduate School of Engineering, the University of Tokyo, through the Doctoral Student Special Incentives Program (SEUT-RA). We wish to thank Dr. Wuquan Li for his invaluable suggestions.

## Author contributions

Le Zhang: Writing – original draft, Visualization, Validation, Methodology, Investigation, Formal analysis, Data curation, Conceptualization. Yunfeng Liang: Writing – review & editing, Validation, Supervision, Methodology, Investigation, Formal analysis, Conceptualization. Arata Kioka: Writing – review & editing, Investigation. Takeshi Tsuji: Writing – review & editing, Supervision, Project administration, Investigation, Funding acquisition, Conceptualization.

## Competing interests

The authors declare no competing interests.

## Ethical approval

This study follows ethical guidelines to ensure inclusivity and transparency.
