## [Transparent Peer Review file · Communications Engineering]

Critical Threshold of 70% CO₂ Concentration for Economically Viable Geological Storage from Direct Air Capture

Corresponding Author: Dr Yunfeng Liang

Version 0:

Reviewer comments:

Reviewer #1

(Remarks to the Author)

The work by Zhang et.al applied MD simulation methods to analyze the influence of impurities on CO₂ storage over a wide range of temperatures and pressures. Their results suggest a strong similarity between the changes in density due impurities at the macroscopic level and the changes in Van der Waals interactions at the molecular level. Furthermore, they proposed normalized storage efficiency factor to be used in real application scenarios, for instance to evaluate the cost. The manuscript is well written, and the study is of interest. One concern I have is the impurity levels considered in this study. Based on my understanding, the phrase direct air capture (DAC) is generally applied to capturing CO₂ from atmosphere, where the CO₂ concentration is much lower. Unless the system captures CO₂ with high selectivity, or high specificity, the impurity levels could be higher, and a discussion of these lower CO₂ levels can be beneficial. Authors can debate this argument. Also, authors mention "Given that a four-stage m-DAC system has already demonstrated a capture 130 efficiency of ~45%, this work focuses on examining the impact of storing impure CO₂ with concentrations ranging from 50 mol% to 100 mol%." Could the authors please direct these studies in their revision. Overall, this work is of interest and would benefit the DAC field in a positive way.

Reviewer #2

(Remarks to the Author)

This work by Zhang and co-workers interestingly shows the effect of density variations in captured CO₂ due to presence of impurities, on the storage efficiency and cost. Using Molecular Dynamics (MD) simulations, this work is successful in demonstrating the variations in the density of captured CO₂ at different storage conditions, i.e. in a range of temperatures and pressures for different concentrations of captured CO₂ from a DAC process. According to the reviewer, this work will contribute significantly to the DAC literature and qualifies for publication in Nature Communications Engineering, with the following comments addressed.

1) The authors have shown significant contributions of O₂ and N₂ impurities on the captured CO₂ density. Typically, a DAC process involves co-adsorption of H₂O along with CO₂, which would make H₂O a major contributing impurity as well. The authors are requested by the reviewer to include an explanation on the influence of H₂O on the density of captured CO₂ and effectively on the storage efficiency. The authors are also requested to clarify the reason for not including H₂O along with O₂ and N₂ for their simulations.

2) The authors have used extensive MD simulations to evaluate captured CO₂ densities with different impurity levels in a wide range of temperature and pressure conditions spanning several different DAC sites. The authors validate their simulation results with reports in NIST Webbook. However, the reviewer feels that the authors should additionally compare with density calculated under varying pressures and temperatures for different impurity levels using equation of state calculations as additional validation for their MD simulation results.

3) The authors talk about how the Normalized Storage Efficiency caused by impurities (NSEI) can affect storage, pressurization and transportation. However, the captured CO₂ can also be further utilized by conversion of CO₂ to value-added chemicals/products. The authors are requested to comment on how the NSEI can affect CO₂ conversion capabilities of the captured CO₂.

4) The authors elaborately focus on the storage efficiency and cost in this work, which is surely a valuable addition to DAC literature. However, the reviewer feels that the authors should comment on how the NSEI can ultimately affect the overall cost of DAC (in US\$/tCO₂), with additional explanations indicating the optimum storage condition recommendations for every DAC plant considered in the work to minimize the overall cost.

Reviewer #3

(Remarks to the Author)

Direct air capture (DAC) is a widely recognized negative emission technology that currently faces challenges of high energy consumption. Unlike traditional strategies, this work proposes a novel concept that the energy consumption of DAC can be reduced by producing lower-purity CO₂ (70 mol%). Theoretical analysis demonstrates that storing lower-purity CO₂ will not significantly increase storage costs. The following questions/comments are raised regarding this manuscript:

1. Given the focus of this work on CO₂ storage, excessive discussion about DAC technology status in the Introduction should be reduced. Greater emphasis should be placed on summarizing current advancements in CO₂ storage technologies.
2. Is specifying the CO₂ source as DAC essential to this study? Considering that point-source carbon capture systems also require CO₂ storage solutions, this distinction may warrant further justification.
3. In the Results section, do all referenced CCS projects specifically involve DAC applications? Clarification is needed regarding the relevance of these examples to the DAC context.
4. Considering the absence of experimental validation, how can the reliability of the simulations in Fig. 1 be verified? Does the Widom line methodology adequately ensure the accuracy of MD simulation results?
5. A comparative analysis between the simulation results and existing equation-of-state models would strengthen the methodological validation.
6. What potential effects might the O₂/N₂ ratio have on the outcomes? This critical parameter requires explicit discussion.
7. While the study analyzes CO₂ stream density, this parameter alone cannot fully determine storage energy consumption and cost. Additional thermodynamic and economic parameters should be incorporated for comprehensive analysis.
8. The cost analysis section would benefit from explicitly establishing the relationship between DAC cost and CO₂ purity, rather than discussing general DAC costs.
9. The quality of figures (e.g., Fig. 6) require enhancement to meet publication standards.

Version 1:

Reviewer comments:

Reviewer #1

(Remarks to the Author)

Authors have satisfactorily addressed the comment from last round. The manuscript is recommended for publication.

Reviewer #2

(Remarks to the Author)

According to the reviewer, the authors have addressed all the comments/suggestions by the reviewer adequately. The submission titled "Critical Threshold of 70% CO₂ Concentration for Economically Viable Geological Storage from Direct Air Capture" will be a valuable addition to the overall CO₂ capture literature, specifically adding interesting insights on considerations for CO₂ storage optimization. Hence, the reviewer has no further recommendations and considers this submission ready to proceed for publication.

Reviewer #3

(Remarks to the Author)

The authors have addressed my concerns appropriately.

List of Responses to Reviewers' Comments and Suggestions

Dear Editor,

Thank you very much for your kind consideration and the reviewers' valuable comments regarding our manuscript. We are pleased to note that all three reviewers provided overall positive feedback and sincerely thank them for their constructive and thoughtful comments. In response, we have made careful revisions. The main improvements are summarized below:

1. **Method Validation:** To enhance confidence in our MD simulation results, we added comparisons between MD simulations and Equation-of-State (EoS) calculations (see newly added **Fig. 1**; previously, it was in Supporting Information; the data from EoS was newly calculated and added). These comparisons further confirm the reliability of our methodology.
2. **Impurity Types and Levels:** We provided a detailed explanation of the chosen O₂: N₂ ratio (1:4) and presented density calculations for mixtures with 2.5% O₂ and 2.5% N₂. The results demonstrate that variations in the O₂: N₂ ratio have a negligible impact on CO₂ density. Regarding H₂O and other impurities, we clarified our considerations for excluding them and added relevant discussion in the manuscript.
3. **Introduction:** We revised the Introduction to better frame the motivation and scope of our work. We make the second and third paragraphs more concise.
4. **Discussion:** We improved the visual quality of **Fig. 7** to enhance overall readability.

We have also made edits on formatting according to the formatting guidelines:

- The title was shortened to 15 words.
- The Abstract was reduced to meet the 150-word limit.
- The Methods section was moved from Supplementary Information to the main text.
- Figure 7 was enhanced for publication quality.
- All figures in the supplementary Information were referenced in the main text.

In addition, we addressed all other minor questions and improved our manuscript when appropriate. Revised parts are highlighted in red in our revised manuscript (please see a **Highlighted-in-Red Copy**). A **Clean Copy** is submitted as guided. The point-by-point responses to the reviewers' comments and changes are as follows:

Reviewer(s)' Comments to Author:

Reviewer: 1

Comments: The work by Zhang et.al applied MD simulation methods to analyze the influence of impurities on CO₂ storage over a wide range of temperatures and pressures. Their results suggest a strong similarity between the changes in density due impurities at the macroscopic level and the changes in Van der Waals interactions at the molecular level. Furthermore, they proposed normalized storage efficiency factor to be used in real application scenarios, for instance to evaluate the cost. The manuscript is well written, and the study is of interest. One concern I have is the impurity levels considered in this study. Based on my understanding, the phrase direct air capture (DAC) is generally applied to capturing CO₂ from atmosphere, where the CO₂ concentration is much lower. Unless the system captures CO₂ with high selectivity, or high specificity, the impurity levels could be higher, and a discussion of these lower CO₂ levels can be beneficial. Authors can debate

this argument. Also, authors mention “Given that a four-stage m-DAC system has already demonstrated a capture efficiency of ~45%, this work focuses on examining the impact of storing impure CO₂ with concentrations ranging from 50 mol% to 100 mol%.” Could the authors please direct these studies in their revision. Overall, this work is of interest and would benefit the DAC field in a positive way.

Response: Thanks a lot for your interest and kind recommendation. We agree that for capture systems, high selectivity or specificity is essential to effectively increase CO₂ concentration. Considering that atmospheric CO₂ levels are only about 420 ppm (~ 427 ppm in 2024), a substantial amount of energy is required to achieve high purity, and the associated purification cost remains one of the main barriers to the large-scale deployment of DAC.

From a CO₂ storage perspective, injecting impure CO₂ would increase the storage cost. To store the same mass of CO₂, impure CO₂ occupies a larger volume, further resulting in faster pressure buildup and a higher risk of leakage. Without considering capture cost, injecting pure CO₂ would be the most stable and conservative approach.

This study aims to explore a balance between CO₂ purity and cost, striving to maximize storage efficiency while keeping the cost within acceptable limits. To this end, it sets a target of concentration to be reached in the near future in the capture community, especially for the DAC techniques. On the other hand, it attracts attention in the storage community, for impurity CCS project planning.

Action: We added a few sentences in the Discussion section (Line 356-361) as below:

In summary, this study aims to explore a balance between CO₂ purity and cost, striving to maximize storage efficiency while keeping the cost within acceptable limits. To this end, it sets a target of 70% as the economically viable concentration to be reached in the near future in the capture community, especially for the DAC techniques. A material system with high selectivity or specificity would be desirable to effectively increase CO₂ concentration. On the other hand, it also attracts attention in the storage community, for impurity CCS project planning.

In addition, we wrapped up the discussion on DAC technology in the Results section (Line 133-140) as below:

Among the three main DAC technologies, membrane-based DAC (m-DAC) offers significant energy advantages. It is reported that the CO₂ produced in a four-stage membrane system could reach 42.4 mol%, while also achieving net-negative CO₂ emissions—meaning the CO₂ emitted during the purification process is less than the amount captured. A slight improvement in membrane selectivity may increase the CO₂ concentration in the permeate to 50% or higher¹⁷. This work focuses on examining the impact of storing impure CO₂ with concentrations ranging from 50 mol% to 100 mol%.

This is also because Reviewer 3 suggested reducing the Discussion of DAC technology in the Introduction section. Please see also Reviewer Report 3.1.

Reviewer: 2

Comments: *This work by Zhang and co-workers interestingly shows the effect of density variations in captured CO₂ due to presence of impurities, on the storage efficiency and cost. Using Molecular Dynamics (MD) simulations, this work is successful in demonstrating the variations in the density of captured CO₂ at different storage conditions, i.e. in a range of temperatures and pressures for different concentrations of captured CO₂ from a DAC process. According to the reviewer, this work will contribute significantly to the DAC literature and qualifies for publication in Nature Communications Engineering, with the following comments addressed.*

2.1 *The authors have shown significant contributions of O₂ and N₂ impurities on the captured CO₂ density. Typically, a DAC process involves co-adsorption of H₂O along with CO₂, which would make H₂O a major contributing impurity as well. The authors are requested by the reviewer to include an explanation on the influence of H₂O on the density of captured CO₂ and effectively on the storage efficiency. The authors are also requested to clarify the reason for not including H₂O along with O₂ and N₂ for their simulations.*

Response: Thank you for your insightful comments and feedback. To our knowledge, the presence of water vapor also occupies volume within the mixture, which may reduce the overall CO₂ density. However, in the context of NSEI, water is not limited to DAC applications—meaning that the reference state used for NSEI calculation may also include water. Therefore, it is fair to say that water will not affect the NSEI results statistically.

From the point of storage side, water is also unavoidable, in saline aquifers or any type of subsurface reservoirs. Again, this is not specific to CO₂ storage from DAC only, but for all types of sources.

Action: We added a sentence in Discussion section as a limitation of this study as below:

Line 365-367:

Beyond that, the influences of water, sulfur-, and nitrogen-oxides from the industrial sector on impure CO₂ storage, as well as the potential impact on CO₂ utilization and conversions, still require further investigation. Overall, the information that we learned from this study is essential for DAC capture technologies development and storage project planning.

In addition, we have added an explanation in the Methods section as below:

Line 377-379:

Because the DAC captures CO₂ directly from the air, we considered the N₂ and O₂ as the primary impurities. Water is not considered since it is inevitable in CO₂ storage, which is not specific for CO₂ from DAC only, but for CO₂ from all types of sources. The ratio of N₂ to O₂ in the simulation is kept at 4:1.

2.2 *The authors have used extensive MD simulations to evaluate captured CO₂ densities with different impurity levels in a wide range of temperature and pressure conditions spanning several different DAC sites. The authors validate their simulation results with reports in NIST Webbook. However, the reviewer feels that the authors should additionally compare with density calculated under varying pressures and temperatures for different*

impurity levels using equation of state calculations as additional validation for their MD simulation results.

Response: Thank you for your valuable feedback. In response, we first calculated the density of pure CO₂ using the Peng–Robinson equation of state (P–R EoS) at 325.65 K. The results are shown in Fig. R1. It can be seen that the P–R EoS performs well in the low-pressure region, while in the high-pressure region, the calculated density exhibits noticeable deviations. The acentric factor used in the model is 0.2667 [32]. Besides a good performance at the high-pressure region, MD offers additional advantages that equation-of-state models do not provide, such as the ability to analyze intermolecular interactions at the microscopic level.

We acknowledge that various EoS models may be capable of accurately predicting impure CO₂ densities; however, evaluating and validating different EoS approaches is not the primary focus of this work.

Action: We really appreciate your valuable advice, and an updated validation result was added in the manuscript (Fig. 1) for further validation and enhancing readability.

Fig. R1. (Newly added Fig. 1 in the manuscript) Density calculation validation results for pure CO₂ at 326.65 K.

Newly Added Reference:

[32] Chappellear, P. S. Acentric factor for carbon dioxide. *Fluid Phase Equilib.* **9**, 319–322 (1982).

2.3 The authors talk about how the Normalized Storage Efficiency caused by impurities (NSEI) can affect storage, pressurization and transportation. However, the captured CO₂ can also be further utilized by conversion of CO₂ to value-added chemicals/products. The authors are requested to comment on how the NSEI can affect CO₂ conversion capabilities of the captured CO₂.

Response: Thank you for pointing out another usage of captured CO₂-conversion to value-added chemicals/products. This is another vital part of Carbon Capture Utilization and Storage (CCUS). However, this work mainly focuses on the storage efficiency issue caused

by impure CO₂.

Action: We add a sentence in the Discussion section as a limitation of this study as below:

Line 363-369:

Moreover, there are efforts to utilize CO₂ from DAC, for example, to convert CO₂ to value-added chemicals/products, which are not considered in this study. Beyond that, the influences of water, sulfur-, and nitrogen-oxides from the industrial sector on impure CO₂ storage, as well as the potential impact on CO₂ utilization and conversions, still require further investigation. Overall, the information that we learned from this study is essential for DAC capture technologies development and storage project planning.

2.4 The authors elaborately focus on the storage efficiency and cost in this work, which is surely a valuable addition to DAC literature. However, the reviewer feels that the authors should comment on how the NSEI can ultimately affect the overall cost of DAC (in US\$/tCO₂), with additional explanations indicating the optimum storage condition recommendations for every DAC plant considered in the work to minimize the overall cost.

Response: Thank you for your insightful and valuable considerations. We agree that a robust cost estimation could further enhance the applicability of our evaluation model. However, DAC costs are influenced by a wide range of factors beyond NSEI alone—such as economic conditions and electricity price, and even the type of energy source used. Therefore, estimating the cost based solely on NSEI may not fully capture the overall cost of DAC projects. Given the complexity of real-world scenarios and the uniqueness of each project, we propose using the inverse of NSEI as a practical correction factor for storage costs. For instance, when NSEI is 0.7, the associated storage cost for impure CO₂ would be approximately $1 \div 0.7 \approx 1.43$ times that of pure CO₂. This approach provides a straightforward method to incorporate NSEI into cost evaluations.

Action: We have strengthened the cost-related discussion in the Results section based on your comments.

Line 231-232:

Regarding other capture-related costs, such as pressurization and transportation, an impure system requires more energy, and the associated costs can be expressed as $1/\text{NSEI}$ times those of pure CO₂. For example, when NSEI is 0.7, the associated impure CO₂ storage cost would be approximately $1 \div 0.7 \approx 1.43$ times that of pure CO₂.

Line 243-246:

For example, at 50 mol% concentration, the NSEI for the CarbFix and Cranfield projects approaches 35%, whereas the efficiency for the Sleipner project falls below 15%. From a cost perspective, this implies that under the same 50 mol% condition, the NSEI-derived storage cost at CarbFix and Cranfield is about 2.3 times that at Sleipner. This underscores the sensitivity of NSEI and associated cost to CO₂ concentrations.

Reviewer 3:

Comments:

Direct air capture (DAC) is a widely recognized negative emission technology that currently faces challenges of high energy consumption. Unlike traditional strategies, this work proposes a novel concept that the energy consumption of DAC can be reduced by producing lower-purity CO₂ (70 mol%). Theoretical analysis demonstrates that storing lower-purity CO₂ will not significantly increase storage costs. The following questions/comments are raised regarding this manuscript:

3.1. Given the focus of this work on CO₂ storage, excessive discussion about DAC technology status in the Introduction should be reduced. Greater emphasis should be placed on summarizing current advancements in CO₂ storage technologies.

Response: Thank you for your thoughtful comment pointing out that the section on DAC technology in the Introduction was long. We have carefully shortened the relevant parts and added more current advancements in CO₂ storage technologies in the Introduction.

Action: We made modifications in the Introduction section as follows:

Line 80 – 86 (newly added):

... In contrast, studies on storage efficiency have largely focused on pure CO₂. Bachu reviewed the assessment of CO₂ storage efficiency in deep saline aquifers, highlighting the influence of storage site characteristics, confining aquifers, operating conditions, and regulatory frameworks [22]. Li *et al.* investigated the impact of pressure buildup on storage efficiency during pure CO₂ injection using a “saturation attack” strategy [23]. Kim *et al.* developed an artificial neural network (ANN) model trained on various geological factors and their ranges, significantly improving the prediction accuracy for CO₂ storage performance [24]. ...

Newly Added References

[22] Bachu, S. Review of CO₂ storage efficiency in deep saline aquifers. *Int. J. Greenhouse Gas Control* **40**, 188–202 (2015).

[23] Li, J. et al. Site-Level Carbon Storage Potential Estimation in Offshore China. in SPE Symposium and Exhibition - Production Enhancement and Cost Optimization D021S007R001 (SPE, Kuala Lumpur, Malaysia, 2024). doi:10.2118/220657-MS.

[24] Kim, Y., Jang, H., Kim, J. & Lee, J. Prediction of storage efficiency on CO₂ sequestration in deep saline aquifers using artificial neural network. *Appl. Energy* **185**, 916–928 (2017).

While also reduce discussion about DAC technologies in Introduction:

Line 50-53 (shortened):

Recently, an amine-functionalized COF-999 (Covalent Organic Frameworks, COF) achieved over 80 mol% CO₂ capture efficiency from the air¹³. ~~This material demonstrated full performance retention after more than 100 adsorption-desorption cycles conducted in the open air of Berkeley, California, and operates at a low regeneration temperature of approximately 60 °C.~~ I-DAC employs a selective liquid-phase chemical absorption of atmospheric CO₂, yielding a CO₂ stream with >95 mol% purity¹⁴. However, energy

consumption is still a concern.

Line 55-63:

Although m-DAC is still in its early stages, it consumes only 1/30 to 1/20 of the energy required by amine-based adsorption systems¹⁵. As a purely physical process with no toxic emissions, m-DAC minimizes environmental impact and public concerns. Moreover, in regions with advanced renewable energy infrastructure, integrating renewables with m-DAC further lowers energy consumption¹⁶. Fujikawa *et al.* reported that a four-stage membrane system achieved a CO₂ permeate concentration of 42.4 %¹⁷. New developing membrane materials, such as graphene oxide-supported deep eutectic solvent membranes (GO-SDESMs)¹⁸, also exhibit excellent separation performance. The low atmospheric CO₂ concentration demands high energy input and extra processing¹¹, making cost the main barrier to large-scale DAC deployment.

Line 73-75:

Even in seismically active Japan, over 100 Gt of CO₂ could be stored—enough to cover more than 100 years of the country's emissions²⁰. The relatively low storage costs combined with the vast storage potential make the storage of impure CO₂ a feasible strategy for reducing overall costs.

We found discussion on DAC technology could be useful for impurity CCS project planning and hence re-organized those sentences in the Results section (Line 132-137) as below:

Among the three main DAC technologies, membrane-based DAC (m-DAC) offers significant energy advantages. It is reported that the CO₂ produced in a four-stage membrane system could reach 42.4 mol%, while also achieving net-negative CO₂ emissions—meaning the CO₂ emitted during the purification process is less than the amount captured. A slight improvement in membrane selectivity may increase the CO₂ concentration in the permeate to 50% or higher¹⁶. This work focuses on examining the impact of storing impure CO₂ with concentrations ranging from 50 mol% to 100 mol%.

3.2. Is specifying the CO₂ source as DAC essential to this study? Considering that point-source carbon capture systems also require CO₂ storage solutions, this distinction may warrant further justification.

Response: Thanks for pointing out that point-source emission, we fully agree that point-source such as cement industry, iron and steel industry and power plants are important CO₂ emission sources. In this paper, we primarily focus on Direct Air Capture (DAC) technology, which captures CO₂ directly from the atmosphere. As a complement to point-source emission capture, DAC offers several unique advantages. It provides a means to offset emissions that are challenging to eliminate (like heavy industry and long-distance transport) and can be deployed directly at the storage site, eliminating the need for CO₂ transportation (from point source to storage site). This is particularly beneficial for CO₂ storage in remote areas (such as deserts or onshore), where transportation represents a significant portion of the overall cost. In contrast, point-source emissions typically have much higher CO₂ concentrations, which helps reduce purification costs. However, such sources often contain sulfur and nitrogen oxides, imposing stricter requirements on absorbents due to potential degradation or reduced efficiency. Examining the impact of point-source CO₂ emissions would be included in our next work.

Action: To better highlight these distinctions, we have made the following modification in the Introduction, Results, and Discussion sections:

Line 64-67:

Even under optimistic **point-source capture conditions**, such as capturing directly from industrial chimneys with CO₂ concentrations of ~20%, the energy requirements remain substantial, **—let alone the additional costs associated with adsorbent degradation caused by impurities such as sulfur and nitrogen compounds.**

Line 105:

This paper applies MD simulation methods to analyze the influence of impurities **from DAC** on CO₂ storage over a wide range of temperatures and pressures.

Line 133:

DAC captures low-purity CO₂ directly from the air **or emission sources**, with O₂ and N₂ being the primary impurities.

Line 365-367:

Beyond that, the influences of water, sulfur-, and nitrogen-oxides from the industrial sector on impure CO₂ storage, as well as the potential impact on CO₂ utilization and conversions, still require further investigation. Overall, the information that we learned from this study is essential for DAC capture technologies development and storage project planning.

We believe these modifications made in response to your valuable comments will help readers understand the distinctions between different CO₂ capture sources.

In addition, we have added an explanation in the Methods section as below:

Line 377-379:

Because the DAC captures CO₂ directly from the air, we considered the N₂ and O₂ as the primary impurities. Water is not considered since it is inevitable in CO₂ storage, which is not specific for CO₂ from DAC only, but for CO₂ from all types of sources. The ratio of N₂ to O₂ in the simulation is kept at 4:1.

This enables the system in our study to be more accurately defined. Please see also Reviewer Report 2.1.

3.3. In the Results section, do all referenced CCS projects specifically involve DAC applications? Clarification is needed regarding the relevance of these examples to the DAC context.

Response: Thank you for this valuable suggestion. To our knowledge, the referenced projects in this work do not report direct DAC applications. Instead, these projects were used to help identify a reasonable and broad pressure–temperature range for the MD simulations. Moreover, this study evaluates the feasibility of implementing impure CO₂ storage, which could offer practical guidance for the planning of such storage strategies. We have further added clarification in the Results section to emphasize this relevance.

Action: In the manuscript, we have revised the Results section to be more emphasized:

Line 140–143:

... A survey of several well-known CCS projects worldwide—CarbFix^{32,33}, Cranfield³⁴, Sleipner^{35–37}, Decatur^{38,39}, and Nagaoka^{40,41}, and China Offshore²¹—guided the temperature (20–120 °C) and pressure (60–320 bar) ranges in this study. For details, refer to the Methods section. ...

Line 343–344:

We also conducted a comparison of NSEI variations across referenced projects, which could provide practical guide for implementing impure CO₂ storage in these projects. The variation in NSEI across 6 projects under varying impurity conditions is clearly illustrated in ...

Thanks again for your valuable advice.

3.4. Considering the absence of experimental validation, how can the reliability of the simulations in Fig. 1 be verified? Does the Widom line methodology adequately ensure the accuracy of MD simulation results?

Response: Thank you for your valuable feedback regarding the validation of the method. The molecular models and force fields used in this study have been extensively validated, showing good agreement with experimental data in properties such as density, vapor–liquid equilibrium (VLE) and heat capacity [62–64], which forms the basis for ensuring calculation reliability. Prior to applying these models, we calculated the densities of pure substances and compared them with NIST data, yielding excellent agreement. This part was added to the main text in the Methods section. For the Widom line, the appearance of this line in Fig. 2 further reflects the accuracy of the simulation results.

Action: We moved the Methods section from Supporting Information into the Results section in the manuscript. And an updated validation result including EoS result was added in the manuscript (Fig. 1) for further validation and enhancing readability.

Line 123–128:

At first, the densities of pure CO₂, N₂, and O₂ were calculated to validate our results from MD simulations. As shown in Fig. R2, the simulation results align closely with NIST WebBook data³⁰, with an average error below 1%. Furthermore, pure CO₂ density was also calculated using the Peng–Robinson equation-of-state (P–R EoS) model^{31,32}. The P–R EoS aligns well with NIST data at low pressures but shows deviations at higher pressures. These results confirm the reliability of our simulation methodology and force field for accurate density predictions.

Fig. R2 (Newly added Fig. 1 in the manuscript) Density calculation validation results for pure CO₂ at 326.65 K.

Reference

- [62] Aimoli, C. G., Maginn, E. J. & Abreu, C. R. A. Transport properties of carbon dioxide and methane from molecular dynamics simulations. *J. Chem. Phys.* **141**, 134101 (2014).
 [63] Makimura, D. et al. Application of Molecular Simulations to CO₂-Enhanced Oil Recovery: Phase Equilibria and Interfacial Phenomena. *SPE J.* **18**, 319–330 (2013).
 [64] Yang, X., Feng, Y., Jin, J., Liu, Y. & Cao, B. Molecular dynamics simulation and theoretical study on heat capacities of supercritical H₂O/CO₂ mixtures. *J. Mol. Liquids* **299**, 112133 (2020).

Newly Added Reference

- [30] Lemmon, E. W. Thermophysical Properties of Fluids. *NIST* (2009).
 [31] Peng, D.-Y. & Robinson, D. B. A new two-constant equation of state. *Ind. Eng. Chem. Fund.* **15**, 59–64 (1976).
 [32] Chappellear, P. S. Acentric factor for carbon dioxide. *Fluid Phase Equilib.* **9**, 319–322 (1982).

3.5. A comparative analysis between the simulation results and existing equation-of-state models would strengthen the methodological validation.

Response: Thank you for your helpful and insightful advice. We compared the pure CO₂ density results obtained from the Peng–Robinson equation of state (P–R EoS) model, NIST data, and molecular dynamics (MD) simulations, as shown in Fig. R3. The P–R EoS model shows good agreement with NIST data in the low-pressure region; however, deviations are observed at higher pressures. In the calculations, the acentric factor of CO₂ was set to 0.2667 [32], with a critical temperature (T_p) of 304.13 K and a critical pressure (P_p) of 73.8 bars. The figure in manuscript was also updated with a new one (Fig. 1), including the P-R EoS calculation result for comparison.

Fig. R3 (Newly added Fig. 1 in the manuscript) Density calculation validation results for pure CO₂ at 326.65 K.

We appreciate your insightful suggestion, and this modification help enhance the readability.

Newly Added Reference

[32] Chappellear, P. S. Acentric factor for carbon dioxide. *Fluid Phase Equilib.* **9**, 319–322 (1982).

3.6. What potential effects might the O₂/N₂ ratio have on the outcomes? This critical parameter requires explicit discussion.

Response: Thank you for your comments on the O₂/N₂ ratio, which is one of the key technical aspects of this study. In our current work, the O₂: N₂ ratio is fixed at 1:4 to reflect the actual composition of ambient air. During the DAC process, ambient air is drawn into the capture system, where CO₂ is retained by the capture medium, while N₂ and O₂ are released. As the CO₂ concentration increases in the stream, the relative ratio of N₂ to O₂ should remain consistent with that in air—hence our use of the 1:4 ratio. This approach has also been adopted in other studies, such as the work by Tsuji et al [29]., which used the same ratio to investigate impure CO₂ storage. Zhou et al. also adopted this ratio for mimic the ambient air condition for performing dynamic sorption experiment [13].

To further clarify the potential effects of the O₂/N₂ ratio, we calculated a density diagram at 2.5 mol% impurity + 97.5 mol% CO₂, as shown in Fig. R4. The results indicate that whether the impurity is entirely N₂ or entirely O₂, the CO₂ density remains nearly unchanged, demonstrating that the O₂/N₂ ratio does not significantly impact the results. This is because the critical temperatures of both O₂ (153.5 K) and N₂ (126.5 K) differ greatly from that of CO₂ (306.2 K), and thus their influence on CO₂ behavior in the mixture is minimal. We author thank you for your questions on this critical technical problem.

Fig. R4 Fig Density of 2.5% impurity + 97.5% CO₂

Reference

- [13] Zhou, Z. et al. Carbon dioxide capture from open air using covalent organic frameworks. *Nature* **635**, 96–101 (2024).
- [29] Tsuji, T., Sorai, M., Shiga, M., Fujikawa, S. & Kunitake, T. Geological storage of CO₂–N₂–O₂ mixtures produced by membrane-based direct air capture (DAC). *Greenh. Gases* **11**, 610–618 (2021).

3.7. While the study analyzes CO₂ stream density, this parameter alone cannot fully determine storage energy consumption and cost. Additional thermodynamic and economic parameters should be incorporated for comprehensive analysis.

Response: Thank you for your constructive feedback. We appreciate your highlighting of other factors that influence overall energy consumption. We fully agree that CO₂ stream density alone is not sufficient to determine the overall cost. Thermodynamic parameters, such as the Joule–Thomson effect, as well as economic factors, including the injection rate, can have a significant impact on the feasibility and performance of the entire project.

Action: we have added relevant discussion and recommendations on these aspects in the Discussion section:

Line 361-363:

When implementing impure CCS projects, more factors—such as engineering and additional thermodynamics parameters—should be incorporated for a well-informed decision.

Thanks for your careful considerations.

3.8. The cost analysis section would benefit from explicitly establishing the relationship between DAC cost and CO₂ purity, rather than discussing general DAC costs.

Response: Thank you for this insightful suggestion. We fully agree that establishing a clearer relationship between DAC cost and NSEI would enhance the practical applicability of cost analysis. However, we also recognize that DAC cost is influenced by a wide range of factors beyond NSEI alone—such as thermodynamic and economic parameters, as well as the type of energy source used. For example, in countries with advanced renewable energy infrastructure, integrating renewable energy with m-DAC systems can further reduce energy

consumption [16]. Given the complexity of real applications and the variation across DAC projects, we propose a simplified approach where the storage cost is expressed as the inverse of NSEI. For instance, when NSEI is 0.7, the associated impure CO₂ storage cost would be approximately $1 \div 0.7 \approx 1.43$ times that of pure CO₂. This method offers a straightforward and practical way to incorporate impurity effects into engineering calculations. We sincerely appreciate your valuable feedback.

Action: We have strengthened the cost-related discussion in the Results section and revised the Subtitle of the third part accordingly.

Line 231-232:

Regarding other capture-related costs, such as pressurization and transportation, an impure system requires more energy, and the associated costs can be expressed as $1/\text{NSEI}$ times those of pure CO₂. **For example, when NSEI is 0.7, the associated impure CO₂ storage cost would be approximately $1 \div 0.7 \approx 1.43$ times that of pure CO₂.**

Line 243-246:

For example, at 50 mol% concentration, the NSEI for the CarbFix and Cranfield projects approaches 35%, whereas the efficiency for the Sleipner project falls below 15%. **From a cost perspective, this implies that under the same 50 mol% condition, the NSEI-derived storage cost at CarbFix and Cranfield is about 2.3 times that at Sleipner.** This underscores the sensitivity of NSEI and associated cost to CO₂ concentrations.

Line 252:

Effective Storage **Based on** Cost Analysis

Reference

[16] Gutsch, M. & Leker, J. Co-assessment of costs and environmental impacts for off-grid direct air carbon capture and storage systems. *Commun. Eng.* **3**, 14 (2024).

3.9. The quality of figures (e.g., Fig. 6) require enhancement to meet publication standards.

Response: Thank you for your careful review. We have revised the format of Fig. 6 (now **Fig. 7**), as shown in Fig. R 5. Further improvements will be made in accordance with the publication guidelines and formatting criteria. We sincerely appreciate your valuable suggestions, which have greatly contributed to enhancing the quality of this work.

Action: We modified **Fig. 7**, and more modifications will be made according to the publication guidelines and formatting criteria.

Fig. R 5 (Improved Fig. 7 in the manuscript) NSEI for reference projects

Thank you very much for your valuable comments and advice, which have greatly improved the quality of this paper. We sincerely hope that our work will contribute meaningfully to the community.

We shall look forward to hearing from you.

Yours sincerely,
Yunfeng Liang, the University of Tokyo

Responses to Editorial Request

Dear Editor,

Thank you very much for your kind consideration and the valuable comments and insightful advice from all reviewers. Those valuable suggestions have undoubtedly enhanced the quality of our work. We are glad to see that all the comments from the reviewers have been addressed!

As requested, we have revised our paper to comply with the journal's policies and formatting style. Revised parts are highlighted in red in our revised manuscript (please see a **Highlighted-in-Red** Copy). A **Clean Copy** is submitted as guided. The editorial requests and Reviewers' comments are copied as follows:

EDITORIAL REQUESTS:

** Your manuscript should comply with our policies and format requirements, detailed in our style and formatting guide (<https://www.nature.com/documents/commsj-phys-style-formatting-guide-accept.pdf>).*

Done.

** Please edit your manuscript according to the editorial requests in the attached table, and outline revisions made in the right hand column. If you have any questions or concerns about any of our requests, please do not hesitate to contact me. It is important that each request be addressed in order to avoid delays in accepting your manuscript. Please upload the completed table with your manuscript files.*

Done.

** Please refer to our checklist for a full list of the files that must be provided upon resubmission: <https://www.nature.com/documents/commsj-file-checklist.pdf>. This checklist does NOT need to be submitted with your final documents, it is for your reference only.*

Done.

Reviewer(s)' Comments to Author:

Reviewer: 1

Comments: *Authors have satisfactorily addressed the comment from last round. The manuscript is recommended for publication.*

Response: We appreciate your kind recommendations.

Reviewer: 2

Comments: *According to the reviewer, the authors have addressed all the comments/suggestions by the reviewer adequately. The submission titled "Critical Threshold of 70% CO₂ Concentration for Economically Viable Geological Storage from Direct Air Capture" will be a valuable addition to the overall CO₂ capture literature, specifically adding interesting insights on considerations for CO₂ storage optimization. Hence, the reviewer has no further recommendations and considers this submission ready to proceed for publication.*

Response: Thank you for your kind words and feedback. We sincerely hope that our work will contribute to the community.

Reviewer 3:

Comments: *The authors have addressed my concerns appropriately.:*

Response: Thanks again for your thoughtful comments and suggestions in the previous report, which greatly improved the quality of our work.

We shall look forward to hearing from you.

Yours sincerely,
Yunfeng Liang, PhD
The University of Tokyo